# Adult Body Height Is Associated with the Risk of Type 2 but Not Type 1 Diabetes Mellitus: A Retrospective Cohort Study of 783,029 Individuals in Germany

**DOI:** 10.3390/jcm12062199

**Published:** 2023-03-12

**Authors:** Sven H. Loosen, Sarah Krieg, Andreas Krieg, Tom Luedde, Karel Kostev, Christoph Roderburg

**Affiliations:** 1Department of Gastroenterology, Hepatology and Infectious Diseases, University Hospital Duesseldorf, Medical Faculty of Heinrich Heine University Duesseldorf, 40225 Duesseldorf, Germanychristoph.roderburg@med.uni-duesseldorf.de (C.R.); 2Epidemiology, IQVIA, 60549 Frankfurt, Germany; 3Department of Surgery (A), University Hospital Duesseldorf, Medical Faculty of Heinrich Heine University Duesseldorf, 40225 Duesseldorf, Germany

**Keywords:** T2D, T1D, HbA1c, tall, short, insulin, epidemiology

## Abstract

Background: Diabetes mellitus is a major global health burden associated with high morbidity and mortality. Although a short adult body height has been associated with increased risk of type 2 diabetes (T2D), there are large inconsistencies between the studies. Therefore, we aimed to investigate the association between body height and T2D in a large cohort of adult outpatients in Germany. Methods: A total of 783,029 adult outpatients with available body height data from the Disease Analyzer (IQVIA) database were included in Germany between 2010 and 2020. The incidence of diabetes mellitus (type 1 and type 2) was evaluated as a function of the patients’ body height stratified by age, sex, and body-mass-index (BMI). Results: In both women and men in all age groups, incidence of T2D decreased with the increasing body height (<50, 51–60, 61–70, and >70 years). There was no association between the body height and the individual HbA1c value. In multivariable Cox regression analyses adjusted for patient age and BMI, hazard ratios for the development of T2D were 1.15 (95% CI: 1.13–1.17) for each 10 cm decrease in body height in women and 1.10 (95% CI: 1.09–1.12) in men. No significant association was found between body height and the development of T1D. Conclusions: We present the first data from a large cohort of outpatients in Germany, providing strong evidence for an association between adult body height and T2D. These data add to the current literature and might help in implementing body height into existing diabetes risk stratification tools to further reduce morbidity and mortality worldwide.

## 1. Introduction

Diabetes mellitus is one of the most common metabolic diseases in the world, with a dramatic increase in prevalence and a major economic and public health burden. In 2021, according to the 10th edition of the International Diabetes Federation’s Diabetes Atlas, 537 million people worldwide had diabetes, accounting for one in ten adults with diabetes [1]. It is estimated that the total number of people with diabetes will continue to increase by 46% to 643 million by 2030 and as many as 783 million by 2045 [1]. However, since only about 50% of people with diabetes are aware of their disease [2,3,4], identification of risk factors for diabetes and preventive health measures to detect and diagnose diabetes are important to initiate early treatment and prevent or delay the development of microvascular and macrovascular complications [5]. The etiology of T2D is complex and several risk factors have been identified in recent years [6]. Some of them are modifiable (overweight (BMI 25 kg/m^2^) and obesity (BMI > 30 mg/m^2^), unhealthy diet, hypertension, smoking, physical inactivity, and sedentary behavior), while others are not (age, gender, family history of diabetes, race/ethnic background, and socioeconomic status) [1]. Interestingly, body height has recently been suggested as a potential risk factor for T2D in various populations [7,8,9,10,11,12,13] and is already being considered in some clinical models for predicting T2D [14,15].

A newly published meta-analysis including 25 studies, 16 (9 cross-sectional studies and 7 cohort studies; 261,496 individuals) demonstrated that both in men and women a shorter body height in adulthood may indicate an increased risk for T2D, but also concluded that there are large inconsistencies across studies and that the thresholds at which the risks changed cannot be defined from the available data [16], highlighting the need for further research in the field.

Adult body height is mainly determined by genetics, but factors, such as nutrition, childhood disease burden, socioeconomic conditions, and geographical location, among others, may also affect individuals’ attained height [17,18]. Height matters for health; it is related to mortality and a range of diseases, such as cancers (generally positive associations, e.g., [19]) and cardiovascular diseases (generally inverse associations, e.g., [20]). On the other hand, at least in Western countries, the average height has been steadily increasing, probably as a result of improved living conditions and lifestyles [21,22].

The purpose of this study was to investigate the association between adult body height and the occurrences of both type 1 (T1D) and T2D in a large cohort of over 700,000 outpatients in Germany. 

## 2. Materials and Methods

### 2.1. Database

The present study is based on data from the Disease Analyzer (DA) database (IQVIA). This database contains demographic, diagnostic, and prescription data from patients followed in practices of general practitioners (GP) in Germany. Practices are selected based on multiple factors (i.e., physician’s age, specialty group, community size category, and German federal state). The database is composed of 3–5% of all GP practices in Germany. Diagnosis and prescription data are coded using the International Classification of Diseases, 10th revision (ICD-10), and the Anatomical Classification of Pharmaceutical Products of the European Pharmaceutical Marketing Research Association (EphMRA), respectively. Data are anonymously transferred to IQVIA on a regular basis, and the quality of these data is assessed using several criteria, such as completeness of documentation and linkage between diagnoses and prescriptions. The DA database has been shown to be representative for general and specialized practices in Germany [23] and was previously published in several studies focusing on body height [24,25] as well as diabetes mellitus [26,27].

### 2.2. Study Population 

In this retrospective cohort study, body height values were available for 941,558 (15.0%) of 6,295,118 individuals >18 years (age at index date) followed in the 816 GP practices in Germany between January 2010 and December 2020. The only inclusion criteria was at least one documented body height value. The first body height value documented between January 2010 and December 2020 was considered as the index date. Individuals with a preexisting diagnosis of diabetes (ICD-10: E10-E14) prior to or at the index date were excluded. A total of 783,029 individuals were eventually available for analyses.

### 2.3. Study Outcomes and Variables 

The outcome of the study was the incidence of diabetes diagnoses within the study period as a function of body height. Incidence was calculated as incidence density: The number of observed diabetes cases divided by the patient years at risk.

Individuals were followed until a diagnosis of diabetes, the last follow-up visit, or the end of the study period (31 May 2022). Documented body height was measured by physicians and not self-documented. Body height was included as a four-category variable; for women: ≤160 cm, 161–170 cm, 171–180 cm, >180 cm, and for men: ≤165 cm, 166–175 cm, 176–185 cm, >185 cm. Diabetes diagnoses were analyzed separately for T1D (ICD-10: E10) and T2D (ICD-10: E11).

### 2.4. Statistical Analyses

Age at first visit was compared between body height categories using the analysis of variance (ANOVA). As there was a strong relationship between body height and age (taller people were younger, *p* < 0.001), all analyses were performed by age group or adjusted for age. First, incidence of diabetes by age group is shown for men and women. 

HbA1c values documented within 6 months prior to or at the first diagnosis of T2D were estimated for each body height category separately for women and men. HbA1c values are documented in the database used in % units. We additionally converted HbA1c values from % to mmol/mol using the formula HbA1c (mmol/mol) = 10.929 * (A1c (%)—2.15). The association between body height and diabetes by sex is analyzed with Cox regression models adjusted for age and BMI at the index date. The proportional-hazard assumption was tested by adding the interactions of the body height variables in the model with the logarithm of event time. 

The results of the Cox regression models are displayed as hazard ratios (HR) and 95% CI for each subtype. The HRs express how the risk of diabetes changes by each 10 cm increase in body height. *P*-values lower than 0.05 were considered as statistically significant. All analyses were conducted with SAS 9.4 (SAS Institute, Cary, NC, USA).

## 3. Results

### 3.1. Study Cohort Characteristics

Of 783,029 study patients, 423,203 (54.1%) patients were female with a mean age of 50.1 years and mean body height of 165.1 cm. Male patients (n = 359,826) had a mean age of 48.9 years and a mean body height of 178.5 cm. Among women, most patients (n = 224,741) were categorized in the body height group of 161–170 cm. The majority of men (n = 170,740) were categorized in the body height group of 176–185 cm. Table 1 provides a detailed overview on the study cohort’s characteristics. 

### 3.2. Incidence of T1D and T2D among Patients of Different Body Height Categories

Among women in the age group ≤50 years, the incidence of T2D decreased with the increasing body height from 9.6 cases per 1000 patient years in patients smaller than 160 cm to only 5.8 cases per 1000 patient years in females taller than 180 cm (Figure 1). Similar results were obtained for the other age groups. In the age groups between 51 and 60 years, 61 to 70 years, and >70 years, T2D incidence rates decreased from 20.5 (≤160 cm) to 12.4 (>180 cm), 25.5 to 19.1, and 30.2 to 26.4 cases per 1000 patient years, respectively (Figure 1). Among men, the incidence of T2D decreased from 13.1 (≤165 cm) to 8.8 cases per 1000 patient years (>185 cm) in the age group ≤50 years, from 35.9 (≤165 cm) to 22.2 cases per 1000 patient years (>185 cm) in the age group 51–60 years, from 40.6 (≤165 cm) to 28.7 cases per 1000 patient years (>185 cm) in the age group 61–70 years, and from 39.5 (≤165 cm) to 32.4 cases per 1000 patient years (>185 cm) in the age group >70 years, respectively (Figure 2). Based on the large sample sizes available, all 95% confidence intervals (CI) for these incidence values were very narrow (up to maximal 0.1).

HbA1c values documented within 6 months prior to or at the first diagnosis of T2D did not differ depending on body height. Average HbA1c values were 7.8% in women with a body height of ≤160 cm and 7.8% in women >180 cm. In men, the average HbA1c values were 8.0% in body height category ≤165 cm and 8.2 for patients with a body height of >185 cm (Table 2).

For T1D analyzed in patients ≤40 years only, there was no clear trend of a decreasing incidence in taller patients, but rather a decrease in the incidence among smaller females (Table 3).

### 3.3. Association between the Body Height and the Risk for T1D and T2D

To exclude potential confounders on the observed association between body height and T2D incidence, we next performed multivariate Cox regression analyses adjusted for age and the patients’ body-mass-index (BMI). Here, the HR for the development of T2D was 1.15 (95% CI: 1.13–1.17, *p* < 0.001) for every 10 cm decrease in body height among women and 1.10 (95% CI: 1.09–1.12, *p* < 0.001) among men (Table 4). In contrast, there was no significant association between the body height and the development of T1D for every 10 cm decrease in body height among women (HR: 1.03 (95% CI: 0.87–1.23; *p* = 0.711) or men (HR: 1.06 (95% CI: 0.82–1.22; *p* = 0.779, Table 3).

## 4. Discussion

In this study, we evaluated the association between the body height and diabetes mellitus stratified by sex in a large cohort of over 700,000 adult outpatients in Germany. In line with previous studies [7,28,29,30], we showed that the individual body height is linked with the occurrence of T2D. For both women and men and in each age group, there was a decrease in the incidence of T2D with the increasing body height. Using multivariate Cox regression analyses adjusted for patients’ age and BMI, an HR for the development of T2D of 1.15 was demonstrated for each 10 cm decrease in body height in women and 1.10 in men. In contrast, no significant association between the body height and the development of T1D was found. Previously, Wittenbecher et al. examined data from more than 2600 participants in the European Prospective Investigation into Cancer and Nutrition (EPIC) study, which involved a total of more than 27,500 people [30]. After adjusting for age, potential lifestyle confounders, education, and waist circumference, the authors showed that men and women were more than 30% less likely to develop T2D for every 10 cm difference in body height [30]. Specifically, leg length was inversely associated with diabetes risk [30]. Similar results were obtained by the Shanghai Women’s Health Study and the Shanghai Men’s Health Study, although here, in contrast to the results of our study, the association decreased completely when adjusted for BMI [31].

Although the mechanisms of how body height relates to diabetes risk are largely unknown, several factors have been discussed as possible contributors to this association. For instance, taller individuals have been described to have lower insulin resistance and better beta cell function than shorter individuals [11,30,32,33], which has been attributed in part to lower ectopic fat storage, e.g., in the liver [34]. Interestingly, in a review article, Stefan et al. showed the extent to which non-alcoholic fatty liver disease (NAFLD) and T2D co-occur worldwide. They found that 25% of adults have NAFLD and that the prevalence increases to a remarkable 60% when obesity and/or diabetes are co-existing. The explanation for this association is thought to be that the interaction of fatty liver and diabetes, which includes subclinical inflammation, insulin resistance, elevated glucose levels, dysregulated liver proteins (hepatokines), dyslipidemia, and hypercoagulation of the blood, amplifies the effects of the two diseases on each other [35]. To our knowledge, few studies have examined the association between body height and NAFLD. In this regard, we refer to a large prospective cohort study of 35,994 adults from China, which showed that larger body height was inversely associated with the risk of NAFLD in both men and women [36]. However, further studies are needed to clarify the relationship between NAFLD and diabetes as a function of body height, including, in particular, experimental studies at the molecular pathological level.

While insulin resistance and impaired insulin secretion are the two main pathophysiological mechanisms for the development of T2D [37], T1D is characterized by absolute insulin deficiency due to progressive destruction of insulin-producing pancreatic beta cells [38,39,40]. These different mechanisms may be at least partially causative of why we found a significant association between body height and T2D, but not T1D in our study.

There are a number of cardiovascular diseases and several cardiometabolic risk factors that have been reported to be associated with lower risk of diabetes in larger individuals, including blood pressure, triacylglycerols, CRP levels, and adiponectin [22,30,41]. Therefore, multiple, overlapping, and complex biological pathways could influence body height and the risk of T2D through an impact on adipose metabolism and adipose tissue function [30]. The association between body height and the development of T2D is also hypothesized to be influenced by other hormonal factors relevant to growth, as well as the intrauterine environment, childhood nutrition, and vitamin D deficiency [42,43,44]. In this context, growth during puberty and increased body height in adulthood have been linked to insulin-like growth factor concentrations [30,45,46], which have been reported to contribute to insulin sensitivity [30,47]. 

Interestingly, our study did not find a correlation between the body height and individual HbA1c levels. In contrast, Bonfig et al. [48], who studied body height at the onset of T1D and height growth as a function of diabetes duration and metabolic control in a large cohort of children and adolescents in Germany and Austria, demonstrated that body height in adulthood correlated with diabetes duration and mean HbA1c, even with intensive insulin therapy and acceptable metabolic control. Whereas in patients with a mean HbA1c value of <7.0%, adult body height was within the normal range, a significant decrease was observed in patients with moderate (mean HbA1c value of 7.0–8.0%) or poor diabetes control (mean HbA1c value of >8.0%). In particular, body height was found to be above average at the time of T1D diagnosis, which was attributed in part to hormonal changes during the prediabetic phase [48,49].

Regarding the question whether metabolic control also influences height growth in T1D, inconsistent results are found in the literature [50,51,52]. Although historical reports on the body height of individual children with T1D indicate a pronounced short stature [53,54], more recent studies have shown at most a slight reduction in body height in T1D compared with individuals without diabetes. The small or absent differences in body height between patients with T1D and healthy individuals may be due in part to medical and technological improvements in T1D over the past century, such as intensified insulin therapy already at the time of diagnosis [48,55].

However, some limitations should be considered when interpreting the study’s results. First, our study is subject to the inevitable limitations of a longitudinal and retrospective analysis of a large database. It should be noted that our data are descriptive only. Secondary data analyses, such as the present study are usually limited by the incompleteness of the underlying data. All diagnoses were documented with ICD-10 codes, which may have led to misclassification and undercoding of certain diagnoses. Height measurements or other covariates, such as BMI were not available in all patients. Therefore, it remains unclear whether this may have led to some selection bias. Our study specifically examined the association between body height and diabetes, which was adjusted for age, sex, and BMI. Certain information that would have allowed for further analysis, such as socioeconomic status, family history, ethnic background/race, environmental conditions, lifestyle factors (e.g., physical activity, nicotine or alcohol use, diet), or other factors associated with increased diabetes risk, such as blood lipid levels, were not available. Furthermore, separate analyses of waist and hip circumference were not performed in our study due to the lack of more detailed information. These parameters could have potentially had an impact on our results, as disproportionate body fat distribution is considered as a risk factor for impaired cardiometabolic health [56]. It should also be noted that our study did not investigate diabetes duration due to the study design. Finally, although the results of our study are supported by the results of previous studies [7,28,29,30], no causal relationships but only associations could be drawn. The strength of our study is, however, the large number of patients included and the use of representative population-based data from Germany. Since the range of variation in body height in a given population is usually small, a large number of events is required for reliable risk assessment. The IQVIA DA database used for the analyses in this study has already been validated by several studies [23] and previously used in studies on body height [24,25] and diabetes mellitus [26,27].

## 5. Conclusions

Overall, our results provide strong evidence that tall individuals have a lower risk of T2D than short individuals. The results of our study may be useful in considering the inclusion of body height in risk stratification tools to predict and assess diabetes risk. However, it remains unclear to what extent the risk of T2D is directly influenced by body height or only indirectly, e.g., via liver fat content or other cardiometabolic risk factors as important influencing variables. Therefore, further clinical and experimental studies are needed to investigate growth factors during pregnancy, early childhood, puberty, and early adulthood in relation to diabetes prevention.

## Figures and Tables

**Figure 1 jcm-12-02199-f001:**
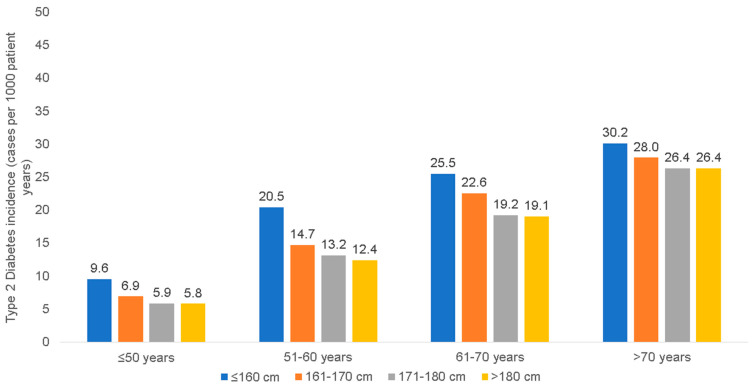
Incidence of type 2 diabetes by age and body height among women.

**Figure 2 jcm-12-02199-f002:**
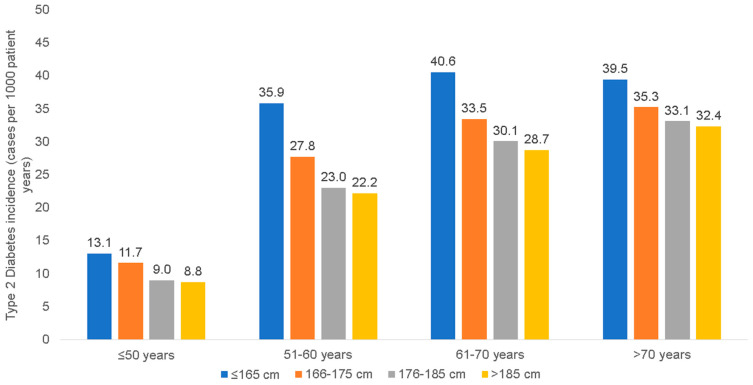
Incidence of type 2 diabetes by age and body height among men.

**Table 1 jcm-12-02199-t001:** Mean age, BMI, and body height at index date.

Variables	Women				
	Total	≤160 cm	161–170 cm	171–180 cm	>180 cm
N	423,203	115,621	224,741	77,995	4846
Age at index date (mean, SD)	50.1 (18.0)	56.4 (18.8)	49.3 (17.4)	43.4 /15.3)	39.6 (13.2)
BMI (mean, SD)	26.3 (6.7)	27.1 (6.2)	26.2 (7.2)	25.5 (5.9)	25.5 (5.9)
Height (mean, SD)	165.1 (6.9)	156.8 (3.3)	165.8 (2.7)	174.3 (2.6)	183.4 (2.9)
Weight (mean, SD)	64.3	66.6	72.0	77.8	85.8
	**Men**				
	**Total**	**≤165 cm**	**166–175 cm**	**176–185 cm**	**>185 cm**
N	359,826	16,231	113,475	170,740	59,380
Age at index date (mean, SD)	48.9 (17.3)	58.8 (19.4)	53.1 (18.0)	47.5 (16.4)	42.4 (14.4)
BMI (mean, SD)	27.3 (5.0)	27.5 (5.3)	27.5 (4.9)	27.3 (5.0)	27.1 (5.0)
Height (mean, SD)	178.2 (7.5)	162.3 (3.3)	171.7 (2.6)	180.2 (2.8)	189.6 (3.4)
Weight (mean, SD)					

**Table 2 jcm-12-02199-t002:** Comparison of HbA1c values between different body height categories for men and female patients.

	HbA1c Value (%)		HbA1c Value (mmol/mol)	
Body Height Category	Women	Men	Women	Men
≤165 cm	7.9	8.2	63	66
166–175 cm	8.0	8.2	64	66
176–185 cm	8.5	8.3	69	67
>185 cm	7.8	8.3	62	67

**Table 3 jcm-12-02199-t003:** Incidence of type 1 diabetes among patients ≤40 years for different body height categories.

	Cases per 1000 Patient Years (95% CI)	
Body Height Category	Women	Men
≤160 cm	0.12 (0.10–0.14)	0.13 (0.11–0.15)
161–170 cm	0.12 (0.10–0.14)	0.10 (0.08–0.12)
171–180 cm	0.19 (0.17–0.21)	0.13 (0.11–0.15)
>180 cm	0.34 (0.32–0.36)	0.18 (0.16–0.20)

**Table 4 jcm-12-02199-t004:** Association between adult body height and diabetes by sex (Cox regression models).

	Women		Men	
	Risk Increase for Every 10 cm Reduction in Body Height (HR, 95% CI)	*p*-Value	Risk Increase for Every 10 cm Reduction in Body Height (HR, 95% CI)	*p*-Value
Type 1 diabetes	1.03 (0.87–1.23)	0.711	1.06 (0.82–1.22)	0.779
Type 2 diabetes	1.15 (1.13–1.17)	<0.001	1.10 (1.09–1.12)	<0.001

## Data Availability

The datasets used and/or analyzed during the current study are available from the corresponding author on reasonable request.

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
