# Peer review of "Adult Body Height Is Associated with the Risk of Type 2 but Not Type 1 Diabetes Mellitus: A Retrospective Cohort Study of 783,029 Individuals in Germany"

_jcm, 2023, doi:10.3390/jcm12062199_

Round 1

Reviewer 1 Report

Dear author.

I appreciate the opportunity to evaluate the manuscript “Adult body height determines the risk of type 2 diabetes mellitus: a retrospective cohort-study of 783,029 individuals in Ger-many”.

I appreciate the opportunity to evaluate the manuscript “Adult body height determines the risk of type 2 diabetes mellitus: a retrospective cohort-study of 783,029 individuals in Germany”.

Despite the study covering a theme already well studied in the literature, the sample size in a developed country is a sizeable point.

1. Abstract: you could insert the clarification of obesity according to WHO in the abstract. Are the authors working with incidence or prevalence? There are conceptual differences and the authors do not make this clear, because sometimes they talk about prevalence, sometimes about incidence.

2. Introduction: can be more clear. Metabolic syndrome is a risk factor for other diseases, add. Adding diabetes epidemiology data with the IDF, we already have data from 2021, with more than 500 million diabetics. Among the non-modifiable ones, include smoking, inadequate diet, physical inactivity and sedentary behavior. Clearly show the cohort studies that showed a positive association between height and DM2. Discuss the limitations and divergences of these studies. What is the literature gap to be filled? Authors should increase their justification. The problem is clear.

3. Methods: quote the study design at the beginning of the section; describes the context such as income, population of Pakistan and included the period of data collection in a topic called “Context”. Describe the population clearly, adults aged 18 and over from how many services? Are there two outcome variables? Incidence of DM1 and DM2? This is not clear. Enter all predictor variables and their treatments. The authors do not mention the reference used to stratify the height groups. The authors do not clearly insert the sample (or they worked with the entire population). Be more clear.

“Age at first visit was compared between body height categories. As there was a strong relationship between body height and age (taller people were younger), all anal-yses were performed either by age group or adjusted for age “. Was there a statistical difference? Use ANOVA or other to check.

The cox modeling needs to be described and all output items need to be described in the statistical analysis session. To review.

4. Results: enter the weight in table 1.

Insert in the method how was the incidence of DM calculated, was it cumulative incidence or incidence density? Figure 1 and 2 – add 95% confidence interval, with age difference statistical tests. Where is the incidence of type 1 diabetes?

There was no correlation between body height and the individual HbA1c values documented within six months prior to or at the first diagnosis of T2D. – add the name of the correlation test and a figure.

Average HbA1c values were 62 mmol/mol (7.8%) – insert and define the variable/collection method in the methods.

Table 2 – add 95%CI, standard deviation and different statistic, same thing as table 3. Why not make a small graph like table 2? Why not use 100,000 inhabitants in the incidence density?

The COX model - how was the assumption of proportionality of risks checked? Add the results and whether the model was proportional. Which variables was the model adjusted for? Write, as it is not clear in the text.

5. Discussion: In this study, we evaluated the association between body height and diabetes mellitus stratified by sex, age group and BMI in a large cohort of over 700,000 adult out-patients in Germany. In line with previous studies;

Where are the Cox analyzes stratified by age and BMI?

Conclusion=review, must respond exclusively to the objective. Be careful when using mediators in this non-mediating analysis.

Author Response

Point-by-point response

Reviewer 

Comments and Suggestions for Authors

Dear author.

I appreciate the opportunity to evaluate the manuscript “Adult body height determines the risk of type 2 diabetes mellitus: a retrospective cohort-study of 783,029 individuals in Germany”.

Despite the study covering a theme already well studied in the literature, the sample size in a developed country is a sizeable point.

  1. Abstract: you could insert the clarification of obesity according to WHO in the abstract. Are the authors working with incidence or prevalence? There are conceptual differences and the authors do not make this clear, because sometimes they talk about prevalence, sometimes about incidence.

Response: We changed prevalence into incidence in the Abstract (this was our mistake). In other parts of the manuscript we always use(d) the term incidence. Regarding obesity, we did not investigated obesity in this study.

  1. Introduction: can be more clear. Metabolic syndrome is a risk factor for other diseases, add. Adding diabetes epidemiology data with the IDF, we already have data from 2021, with more than 500 million diabetics. Among the non-modifiable ones, include smoking, inadequate diet, physical inactivity and sedentary behavior. Clearly show the cohort studies that showed a positive association between height and DM2. Discuss the limitations and divergences of these studies. What is the literature gap to be filled? Authors should increase their justification. The problem is clear.

Response: We have revised the Introduction to focus on the research question of our work. We have now deliberately removed the first section of the introduction that deals with the metabolic syndrome, since the focus of our work is less on obesity or the metabolic syndrome and more on body height in the context of diabetes. In retrospect, this section seemed redundant, so we have made this change. In our Introduction, we have also further justified the aim of our work and emphasized the extent to which our work also has clinical relevance, particularly with regard to the identification of risk factors in the context of prevention/screening to reduce diabetes complications.

  1. Methods: quote the study design at the beginning of the section; describes the context such as income, population of Pakistan and included the period of data collection in a topic called “Context”. Describe the population clearly, adults aged 18 and over from how many services? Are there two outcome variables? Incidence of DM1 and DM2? This is not clear. Enter all predictor variables and their treatments. The authors do not mention the reference used to stratify the height groups. The authors do not clearly insert the sample (or they worked with the entire population). Be more clear.

Response: Dear Reviewer, we perfomed the study in Germany using German data, no data from Pakistan. In 2.2 Study population we wrote in the first version “In this retrospective cohort study, body height values were available for 941,558 (15.0%) of 6,295,118 individuals ≥18 years (age at index date) followed in the 816 GP practices in Germany between January 2010 and December 2020”. This sentence contains information on age (≥18), country and time period of the study).

“Age at first visit was compared between body height categories. As there was a strong relationship between body height and age (taller people were younger), all anal-yses were performed either by age group or adjusted for age “. Was there a statistical difference? Use ANOVA or other to check. The cox modeling needs to be described and all output items need to be described in the statistical analysis session. To review.

Response: We used ANOVA to check but the differences are so large that they can be seen without statistical test, and even if statistical test would be not significant, the differences are large enough to adjust for age.  We added the sentence with ANOVA in Methods.

  1. Results: enter the weight in table 1.

Response: We added weight in Table 1

Insert in the method how was the incidence of DM calculated, was it cumulative incidence or incidence density? Figure 1 and 2 – add 95% confidence interval, with age difference statistical tests. Where is the incidence of type 1 diabetes?

Response: We calculated the incidence density. Now we added this information to Methods. Diabetes diagnoses were analyzed separately for T1D (ICD-10: E10) and T2D (ICD-10: E11). There is still the same outcome but two different diabetes types. Regarding 95% CI, we added them for T1D; by T2D , die to very large samples, all CIs are the same like values, for example 13.6 (13.6-1.3.6).  We added this information to Methods.

There was no correlation between body height and the individual HbA1c values documented within six months prior to or at the first diagnosis of T2D. – add the name of the correlation test and a figure.

Response: We did not mean a correlation but only a relaionship. We modified this sentence as well as following sentences: “HbA1c values documented within six months prior to or at the first diagnosis of T2D did not differ depending on body height. Average HbA1c values were 7.8% in women with a body height of ≤160 cm and 7.8% in women >180 cm. In men, the average HbA1c was 8.0% in body height category ≤165 cm and 8.2 for patients with a body height >185 cm (Table 2).”

Average HbA1c values were 62 mmol/mol (7.8%) – insert and define the variable/collection method in the methods.

Response: In the database used all HbA1c values are documented in %; no mmol/mol. In the first version of the manuscript we transferred % into mmol/mol. However we realized that % values are still the standard method to describe HbA1c values and we removed mmol/mol-values in the new version from the text; they however stay in the Table. We added the following text to Methods:  “HbA1c values documented within six months prior to or at the first diagnosis of T2D were estimated for each body height category separately for women and men. HbA1c values are documented in the database used in % units. We additionally converted HbA1c values from % to mmol/mol using the formula HbA1C (mmol/mol) = 10.929 * (A1C(%) - 2.15).”

Table 2 – add 95%CI, standard deviation and different statistic, same thing as table 3. Why not make a small graph like table 2? Why not use 100,000 inhabitants in the incidence density?

Response: We do not have a total number of inhabitants in the databae and cannot calculate the number of diabetes cases found in the database using inhabitants as denominator. What is why we always have to use patient-years as they can be calculated in the database.

Table 3 was created rather than Figure as the incidences are very small (0,13, 0.18) and they are hard to show in the Figure.

The COX model - how was the assumption of proportionality of risks checked? Add the results and whether the model was proportional. Which variables was the model adjusted for? Write, as it is not clear in the text.

Response: The proportional-hazard assumption was tested by adding the interactions of the body height variables in the model with the logarithm of event time. The association between body height and diabetes by sex is analyzed with Cox-regression models adjusted for age and BMI at the index date. We described that in Methods.

  1. Discussion: In this study, we evaluated the association between body height and diabetes mellitus stratified by sex, age group and BMI in a large cohort of over 700,000 adult outpatients in Germany. In line with previous studies; Where are the Cox analyzes stratified by age and BMI?

Response: We are sorry, this sentence was modified.

Conclusion=review, must respond exclusively to the objective. Be careful when using mediators in this non-mediating analysis.

Response: Thank you for your comment. We have made changes to the Conclusion accordingly.

Reviewer 2 Report

The manuscript by Loosen and colleagues assess the predictability of height for diabetes. The manuscript is well written. I believe the topic, even though, has been assessed by many other studies, is interesting. But I have a few concerns that should be addressed.

1) There are very few grammatical mistakes. For example, Results section 4.3. “Here, the The hazard ratio (HR) for…” and “…(HR: 1.03 (95% CI: 0.87- 1.23; p=0.711) or men (HR: 1.06 (95% CI: 0.82-1.22; p=0.779, Table 3).” There is an extra “the” and missing/extra (). Also, Methods are 2.1, 2.2, 3. and 3.1. I think they all should be 2.X values.

2) Please explain the exclusion criteria better. I am not sure if the authors included patients with scoliosis (severe versus mild), body deformities, Marfan syndrome, or other factors associated with height. Were patients with prediabetes or taking medication to lower serum glucose or increase insulin sensitivity included? How would this effect your results? Also, did the authors consider Metabolic syndrome. These individuals would have a predisposition for diabetes.

3) I am not familiar with the IQVIA dataset. Is this dataset complied using complex sampling? If so, was the analysis performed with complex sampling methods?

4) The authors analysis is interestingly; however, I am interested in the older population, in which changes in height, either due to poor diet or lifestyle, could also be a predictor or have an association with diabetes. Please test for changes in height with diabetes risk or at least discussed it in the discussion.

5) How were extremely tall individual (>2 m) affecting the data?

6) In the results, for tables, place the standard deviation in the tables and remove them from the text. It makes it easier to understand.

7) In the discussion (2nd paragraph), the explanation of how height could be associated with NAFLD is interesting, but I believe there may be too much speculation and the authors do not link height to NALFD. How about visceral fat as a factor more than liver fat? Please describe another mechanism.

Author Response

Point-by-point response

Reviewer

Comments and Suggestions for Authors

The manuscript by Loosen and colleagues assess the predictability of height for diabetes. The manuscript is well written. I believe the topic, even though, has been assessed by many other studies, is interesting. But I have a few concerns that should be addressed.

  • There are very few grammatical mistakes. For example, Results section 4.3. “Here, the The hazard ratio (HR) for…” and “…(HR: 1.03 (95% CI: 0.87- 1.23; p=0.711) or men (HR: 1.06 (95% CI: 0.82-1.22; p=0.779, Table 3).” There is an extra “the” and missing/extra (). Also, Methods are 2.1, 2.2, 3. and 3.1. I think they all should be 2.X values.

Response: These mistakes were corrected.

  • Please explain the exclusion criteria better. I am not sure if the authors included patients with scoliosis (severe versus mild), body deformities, Marfan syndrome, or other factors associated with height. Were patients with prediabetes or taking medication to lower serum glucose or increase insulin sensitivity included? How would this effect your results? Also, did the authors consider Metabolic syndrome. These individuals would have a predisposition for diabetes.

Response: In this study, individuals with a preexisting diagnosis of diabetes (ICD-10: E10-E14) prior to or at index date were excluded. Preexisting diabetes diagnosis was the only excluding criterion. Diseases listed by the reviewers (scoliosis, body deformities) are rarely coded by general practitioners. Moreover, such diagnoses usually occur in children and adolescents and cause the reduction of the height in the young age. We however analyzed adult individuals and the outcome was diabetes (type 2) what usually occur among in the second half of the life. Pre-diabetes is not documented as no ICD 10 code is available.

  • I am not familiar with the IQVIA dataset. Is this dataset complied using complex sampling? If so, was the analysis performed with complex sampling methods?

Response: No complex sampling methods; this database contains electronic medical records of all patients visited office based physicians (general practitioners in our study). No special search for patients like in clinical trials is made, but only available electronic medicals records were analyzed.

  • The authors analysis is interestingly; however, I am interested in the older population, in which changes in height, either due to poor diet or lifestyle, could also be a predictor or have an association with diabetes. Please test for changes in height with diabetes risk or at least discussed it in the discussion.

Response: Under limitations, we wrote that no data on dietary or lifestyle factors were available in the database.

  • How were extremely tall individual (>2 m) affecting the data?

Response: In the database we used, the maximum height was 190 cm; due to data protection rules, weight over 190 cm was set up back to 190 cm. However, these individuals represent less than 0.5% of males and less than 0.001% of females.

  • In the results, for tables, place the standard deviation in the tables and remove them from the text. It makes it easier to understand.

Response: We removed standard deviation from the text and only kept in Tables.

  • In the discussion (2nd paragraph), the explanation of how height could be associated with NAFLD is interesting, but I believe there may be too much speculation and the authors do not link height to NAFLD. How about visceral fat as a factor more than liver fat? Please describe another mechanism.

Response: Thank you for your comment. We have taken your suggestions into account and have amended the section by deleting some passages and highlighting the reference to NAFLD / diabetes and body height.